# Robust Nonparametric Regression under Poisoning Attack

## Abstract

This paper studies robust nonparametric regression, in which an adversarial attacker can modify the values of up to $q$ samples from a training dataset of size $N$. Our initial solution is an M-estimator based on Huber loss minimization. Compared with simple kernel regression, i.e. the Nadaraya-Watson estimator, this method can significantly weaken the impact of malicious samples on the regression performance. We provide the convergence rate as well as the corresponding minimax lower bound. The result shows that, with proper bandwidth selection, $\ell_\infty$ error is minimax optimal. The $\ell_2$ error is optimal if $q \lesssim \sqrt{N/\ln^2 N}$, but is suboptimal with larger $q$. The reason is that this estimator is vulnerable if there are many attacked samples concentrating in a small region. To address this issue, we propose a correction method by projecting the initial estimate to the space of Lipschitz functions. The final estimate is nearly minimax optimal for arbitrary $q$, up to a $\ln N$ factor.

## 1 Introduction

In the era of big data, it is common for some samples to be corrupted due to various reasons, such as transmission errors, system malfunctions, malicious attacks, etc. The values of these samples may be altered in any way, rendering many traditional machine learning techniques less effective. Consequently, evaluating the effects of these corrupted samples, and making corresponding robust strategies, have become critical tasks in the research community [1–10].

Among all types of data contamination, adversarial attack is of particular interest in recent years [11–17], in which there exists a malicious adversary who aims at deteriorating our model performance. With this goal, the attacker alters the values of some samples using a carefully designed strategy. Compared with other types of undesired samples, such as accidental errors or noise, adversarial samples are more challenging to deal with, since their values are altered deliberately instead of randomly. Therefore, any learning models that can withstand adversarial attacks should also be resilient to other corruptions.

Adversarial attack can be divided into *poisoning attack* [11–13], which manipulates training samples to damage the model, and *evasion attack* [14–17], which modifies test samples to generate wrong predictions. We focus on poisoning attack here. For classification problems, the labels can only be altered within several discrete values, thus the impact of poisoning samples is relatively limited [11, 18, 19]. However, regression problems are crucially different, since the response variable is continuous and can be altered arbitrarily far away from its ground truth. Without proper handling, even if only a tiny fraction of training samples are attacked, the model performance may drastically deteriorate. Therefore, for regression problems, defense strategies against poisoning attack are crucially needed.

Submitted to 37th Conference on Neural Information Processing Systems (NeurIPS 2023). Do not distribute.

Despite many previous works toward robust regression problems, most of them focus on parametric models [13, 20–22]. For example, there are several robust techniques for linear models, such as M-estimation [23], least median of squares [24], least trimmed squares [25], etc. However, for nonparametric methods such as kernel [26] and k nearest neighbor estimator, defense strategies against poisoning attack still need further exploration [27]. Actually, designing robust techniques is indeed more challenging for nonparametric methods than parametric one. For parametric models, the parameters are estimated using full dataset, while nonparametric methods have to rely on local training data around the query point. Even if the ratio of attacked samples among the whole dataset is small, the local anomaly ratio in the neighborhood of the query point can be large. As a result, the estimated function value at such query point can be totally wrong. Despite such difficulty, in many real scenarios, due to problem complexity or lack of prior knowledge, parametric models are not always available. Therefore, we hope to explore effective schemes to overcome the robustness issue of nonparametric regression.

In this paper, we provide a theoretical study about robust nonparametric regression problem under poisoning attack. In particular, we hope to investigate the theoretical limit of this problem, and design a method to achieve this limit. With this goal, we make the following contributions:

Firstly, we propose and analyze an estimator that minimizes a weighted Huber loss, which can be viewed as a combination of $\ell_1$ and $\ell_2$ loss functions, and thus achieves a tradeoff between consistency and adversarial robustness. It was originally proposed in [28], but to the best of our knowledge, it was not analyzed under adversarial setting. We show the convergence rate of both $\ell_2$ and $\ell_\infty$ risk, under the assumption that the function to estimate is Lipschitz continuous, and the noise is sub-exponential. An interesting finding is that if $q \lesssim \sqrt{N/\ln^2 N}$, in which $q$ is the maximum number of attacked samples, then the convergence rate is not affected by adversarial samples, i.e. the influence of poisoning samples on the overall risk is only up to a constant factor.

Secondly, we provide an information theoretic minimax lower bound, which indicates the underlying limit one can achieve, with respect to $q$ and $N$. The minimax lower bound without adversarial samples can be derived using standard information theoretic methods [29]. Under adversarial attack, the estimation problem is harder, thus the lower bound in [29] may not be tight enough. We design some new techniques to derive a tighter one. The result shows that the initial estimator has optimal $\ell_\infty$ risk. If $q \lesssim \sqrt{N/\ln^2 N}$, then $\ell_2$ risk is also minimax optimal. Nevertheless, for larger $q$, the $\ell_2$ risk is not optimal, indicating that this estimator is still not perfect. We then provide an intuitive explanation of the suboptimality. Instead of attacking some randomly selected training samples, the best strategy for the attacker is to focus their attack within a small region. With this strategy, majority of training samples are altered here, resulting in wrong estimates. A simple remedy is to increase the kernel bandwidth to improve robustness. Nevertheless, this will introduce additional bias in other regions. It turns out that $\ell_\infty$ risk can be made optimal by adjusting the bandwidth, while $\ell_2$ risk is always suboptimal. Actually, the drawback of the initial estimator is that it does not make full use of the continuity of regression function, and thus unable to correct the estimation.

Finally, motivated by the issues of the initial method mentioned above, we propose a corrected estimator. If the attack focuses on a small region, although the initial estimate fails here, the output elsewhere is still reliable. With the assumption that the underlying function is continuous, the value at such region can be inferred using the surrounding values. With such intuition, we propose a nonlinear filtering method, which makes minimal adjustment to the estimated function in $\ell_1$ sense, to make it Lipschitz continuous. The corrected estimate is then proved to be nearly minimax optimal up to only a $\ln N$ factor.

The remainder of this paper is organized as follows. In section 2, we provide the problem statement as well as the initial estimator by Huber loss minimization. The upper bound and the minimax lower bound are shown in section 3. In section 4, we elaborate the corrected estimator, as well as related theoretical analysis. Numerical simulation results are shown in section 5. Finally, we discuss limitations and provide concluding remarks in section 6 and 7, respectively.

## 2  The Initial Estimator

Suppose $\mathbf{X}_1, \ldots, \mathbf{X}_N \in \mathbb{R}^d$ be $N$ independently and identically distributed training samples, generated from a common probability density function (pdf) $f$. For each sample $\mathbf{X}_i$, we can receive a

corresponding label $Y_i$:

$$Y_i = \begin{cases} \eta(\mathbf{X}_i) + W_i & \text{if } i \notin \mathcal{B} \\ \star & \text{otherwise,} \end{cases} \tag{1}$$

in which $\eta : \mathbb{R}^d \rightarrow \mathbb{R}$ is the unknown underlying function that we would like to estimate. $W_i$ is the noise variable. For $i = 1, \ldots, N$, $W_i$ are independent, with zero mean and finite variance. $\mathcal{B}$ is the set of indices of attacked samples. $\star$ means some value determined by the attacker. For each normal sample $\mathbf{X}_i$, the received label is $Y_i = \eta(\mathbf{X}_i) + W_i$. However, if a sample is attacked, then $Y_i$ can be arbitrary value determined by the attacker. The attacker can manipulate up to $q$ samples, thus $|\mathcal{B}| \leq q$.

Our goal is opposite to the attacker. We hope to find an estimate $\hat{\eta}$ that is as close to $\eta$ as possible, while the attacker aims at reducing the estimation accuracy using a carefully designed attack strategy. We consider white-box setting here, in which the attacker has complete access to the ground truth $\eta$, $\mathbf{X}_i$ and $W_i$ for all $i \in \{1, \ldots, N\}$, as well as our estimation algorithm. Under this setting, we hope to design a robust regression method that resists to any attack strategies.

The quality of estimation is evaluated using $\ell_2$ and $\ell_\infty$ loss, which is defined as

$$R_2[\hat{\eta}] = \mathbb{E}\left[\sup_{\mathcal{A}}(\hat{\eta}(\mathbf{X}) - \eta(\mathbf{X}))^2\right], \tag{2}$$

$$R_\infty[\hat{\eta}] = \mathbb{E}\left[\sup_{\mathcal{A}}\sup_{\mathbf{x}}|\hat{\eta}(\mathbf{x}) - \eta(\mathbf{x})|\right], \tag{3}$$

in which $\mathcal{A}$ denotes the attack strategy, $\mathbf{X}$ denotes a random test sample that follows a distribution with pdf $f$. Our analysis can be easily generated to $\ell_p$ loss with arbitrary $p$.

The kernel regression, also called the Nadaraya-Watson estimator [26, 30] is

$$\hat{\eta}_{NW}(\mathbf{x}) = \frac{\sum_{i=1}^N K\left(\frac{\mathbf{x}-\mathbf{X}_i}{h}\right) Y_i}{\sum_{i=1}^N K\left(\frac{\mathbf{x}-\mathbf{X}_i}{h}\right)}, \tag{4}$$

in which $K$ is the Kernel function, $h$ is the bandwidth that will decrease with the increase of sample size $N$. $\hat{\eta}_0(\mathbf{x})$ can be viewed as a weighted average of the labels around $\mathbf{x}$. Without adversarial attack, such estimator converges to $\eta$ [31]. However, (4) fails even if a tiny fraction of samples are attacked. The attacked labels can just set to be sufficiently large. As a result, $\hat{\eta}_0(\mathbf{x})$ could be far away from its truth.

Now we build the estimator based on Huber loss minimization. Similar method was proposed in [28], but to the best of our knowledge, the performance under adversarial setting has not been analyzed. We elaborate this method for completeness and notation consistency. We use $\hat{\eta}_0$ to denote the new estimator, which is designed as following:

$$\hat{\eta}_0(\mathbf{x}) = \arg\min_{|s| \leq M} \sum_{i=1}^N K\left(\frac{\mathbf{x} - \mathbf{X}_i}{h}\right) \phi(Y_i - s), \tag{5}$$

in which tie breaks arbitrarily if the minimum is not unique, and

$$\phi(u) = \begin{cases} u^2 & \text{if } |u| \leq T \\ 2T|u| - T^2 & \text{if } |u| > T \end{cases} \tag{6}$$

is the Huber cost function.

Here we have introduced two new parameters, namely, $M$ and $T$. With $M \rightarrow \infty$ and $T \rightarrow \infty$, function $\phi$ becomes simple square loss, and it is straightforward to show that the resulting estimator (5) reduces to the Nadaraya-Watson estimator(4). $M$ is a constant hyperparameter that does not change with sample size $N$. By restricting $|s| \leq M$, we avoid the estimated value from being too large. It would be better if $M$ is larger than the upper bound of $|\eta(\mathbf{x})|$, so that the estimation is not truncated too much. $T$ balances accuracy and robustness. Smaller $T$ ensures robustness while sacrificing consistency, and vice versa. To achieve better tradeoff, $T$ need to increase with the training sample size $N$. The best rate of the growth of $T$ with respect to $N$ depends on the strength of the tail of the noise distribution. In our theoretical analysis, we will show that under sub-exponential noise, $T \sim \ln N$ is optimal.

We would like to remark that apart from Huber loss minimization, there are other robust mean estimation methods, such as median-of-means (MoM) [32, 33] and trimmed means [34, 35]. However, it is not efficient to generalize these methods to nonparametric regression. For MoM, with up to $q$ corrupted samples, it divides the data into at least $2q + 1$ groups and then calculate the median of the means of values in each group. Under the regression setting, since the distribution of attacked samples is unknown, we have to divide the data into $2q + 1$ groups within the neighborhood of each query point. As a result, the accuracy with $N$ training samples with $q$ contaminated is only comparable to those with $N/(2q + 1)$ clean samples, indicating that the MoM method is ineffective. Trimmed means method has similar problems. The threshold of the trimmed mean need to be set uniformly among the whole support, while the adversarial attack may focus on a small region. As a result, the parameter can not be tuned optimal everywhere. The nonconsistency at attacked region and the inefficiency at relatively cleaner regions are two problems that can not be avoided simultaneously. Consequently, these alternative approaches are less effective than the M-estimator based on Huber loss minimization.

Finally, we comment on the computation of the estimator (5). Note that $\phi$ is convex, therefore the minimization problem in (5) can be solved by gradient descent. The derivative of $\phi$ is

$$\phi'(u) = \left\{ \begin{array}{lll} 2u & \text{if} & |u| \leq T \\ 2T & \text{if} & u > T \\ -2T & \text{if} & u < -T. \end{array} \right. \tag{7}$$

Based on (5) and (7), $s$ can be updated using binary search. Denote $\epsilon$ as the required precision, then the number of iterations for binary search should be $O(\ln(M/\epsilon))$. Therefore, the computational complexity is higher than kernel regression up to a $\ln(M/\epsilon)$ factor.

## 3  Theoretical Analysis

This section proposes the theoretical analysis of the initial estimator (5) under adversarial setting. To begin with, we make some assumptions about the problem.

**Assumption 1.** *(Problem Assumption) there exists a compact set $\mathcal{X}$ and several constants $L$, $\gamma$, $f_m$, $f_M$, $D$, $\alpha$, $\sigma$, such that the pdf $f$ is supported at $\mathcal{X}$, and*

*(a) (Lipschitz continuity) For any $\mathbf{x}_1, \mathbf{x}_2 \in \mathcal{X}$, $|\eta(\mathbf{x}_1) - \eta(\mathbf{x}_2)| \leq L||\mathbf{x}_1 - \mathbf{x}_2||$;*

*(b) (Bounded $f$ and $\eta$) For all $\mathbf{x} \in \mathcal{X}$, $f_m \leq f(\mathbf{x}) \leq f_M$ and $|\eta(\mathbf{x})| \leq M$, in which $M$ is the parameter used in (5);*

*(c) (Corner shape restriction) For all $r < D$, $V(B(\mathbf{x}, r) \cap \mathcal{X}) \geq \alpha v_d r^d$, in which $B(\mathbf{x}, r)$ is the ball centering at $\mathbf{x}$ with radius $r$, $v_d$ is the volume of $d$ dimensional unit ball, which depends on the norm we use;*

*(d) (Sub-exponential noise) The noise $W_i$ is subexponential with parameter $\sigma$,*

$$\mathbb{E}[e^{\lambda W_i}] \leq e^{\frac{1}{2}\sigma^2\lambda^2}, \forall |\lambda| \leq \frac{1}{\sigma}, \tag{8}$$

*for $i = 1, \ldots, N$.*

(a) is a common assumption for smoothness. (b) assumes that the pdf is bounded from both below and above. (c) prevents the shape of the corner of the support from being too sharp. Without assumption (c), the samples around the corner may not be enough, and the attacker can just attack the corner of the support. (d) requires that the noise is sub-exponential. If the noise assumption is weaker, e.g. only requiring the bounded moments of $W_i$ up to some order, then the noise can be disperse. In this case, it will be harder to distinguish adversarial samples from clean samples. More discussions are provided in section 6.

We then make some restrictions on the kernel function $K$.

**Assumption 2.** *(Kernel Assumption) the kernel need to satisfy: (a) $\int K(\mathbf{u})du = 1$; (b)$K(\mathbf{u}) = 0, \forall ||\mathbf{u}|| > 1$; (c) $c_K \leq K(\mathbf{u}) \leq C_K$ for two constants $c_K$ and $C_K$.*

(a) is actually not necessary, since from (5), the estimated value will not change if the kernel function is multiplied by a constant factor. This assumption is only for convenience of proof. (b) and (c)

actually requires that the kernel need to be somewhat close to the uniform function in the unit ball. Intuitively, if the attacker wants to modify the estimate at some $\mathbf{x}$, the best way is to change the response of sample $i$ with large $K((\mathbf{X}_i - \mathbf{x})/h)$, in order to make strong impact on $\hat{\eta}(\mathbf{x})$. To defend against such attack, the upper bound of $K$ should not be too large. Besides, to ensure that clean samples dominate corrupted samples everywhere, the effect of each clean sample on the estimation should not be too small, thus $K$ also need to be bounded from below in its support.

Furthermore, recall that (5) has three parameters, i.e. $h$, $T$ and $M$. We assume that these three parameters satisfy the following conditions.

**Assumption 3.** *(Parameter Assumption)* $h$, $T$, $M$ *need to satisfy (a)*$h > \ln^2 N/N$*; (b)*$T \geq 4Lh + 16\sigma \ln N$*; (c)*$M > \sup_{\mathbf{x} \in \mathcal{X}} |\eta(\mathbf{x})|$*.*

(a) ensures that the number of samples whose distance to $\mathbf{x}$ less than $h$ is not too small. Actually, for a better tradeoff between bias and variance, $h$ need to grow much faster than $\ln^2 N/N$. (b) requires that $T \sim \ln N$. Actually, the optimal growth rate of $T$ depends on the distribution of noise. Recall that in Assumption 1(d), we assume that the distribution of noise is sub-exponential. If we use sub-Gaussian assumption instead, then it is enough for $T \sim \sqrt{\ln N}$. If the noise is further assumed to be bounded, then $T$ can just be set to constant. (c) prevents the estimate from being truncated too much.

The upper bound of $\ell_2$ error is derived under these assumptions. Denote $a \lesssim b$ if $a \leq Cb$ for some constant $C$ that depends only on $L, M, \gamma, f_m, f_M, D, \alpha, \sigma, c_K, C_K$.

**Theorem 1.** *Under Assumption 1, 2 and 3,*

$$\mathbb{E}\left[\sup_{\mathcal{A}} (\hat{\eta}_0(\mathbf{X}) - \eta(\mathbf{X}))^2\right] \lesssim \frac{T^2 q^2}{N^2 h^d} + h^2 + \frac{1}{Nh^d}. \tag{9}$$

The detailed proof of Theorem 1 is shown in section 2 in the supplementary material. From the proof, it can also be observed that the effect of adversarial samples is higher when they concentrate at a small region instead of distributing uniformly over the whole support. Denote $B_h(\mathbf{x})$ as the ball centering at $\mathbf{x}$ with radius $h$. Even if $q/N$ is small, the proportion of attacked samples within $B(\mathbf{x}, h)$ for some $\mathbf{x}$ may be large, which may result in large error at $\mathbf{x}$.

The next theorem shows the bound of $\ell_\infty$ error:

**Theorem 2.** *Under Assumption 1, 2, 3, if $K(\mathbf{u})$ is monotonic decreasing with respect to $\|u\|$, then*

$$\mathbb{E}\left[\sup_{\mathcal{A}}\sup_{\mathbf{x}} |\hat{\eta}_0(\mathbf{x}) - \eta(\mathbf{x})|\right] \lesssim \frac{Tq}{Nh^d} + h + \frac{\ln N}{\sqrt{Nh^d}}. \tag{10}$$

The proof is in section 3 in the supplementary material. We then show the minimax lower bound, which indicates the information theoretic limit of the adversarial nonparametric regression problem. In general, it is impossible to design an estimator with convergence rate faster than the following bound.

**Theorem 3.** *Let $\mathcal{F}$ be the collection of $f, \eta, \mathbb{P}_N$ that satisfy Assumption 1, in which $\mathbb{P}_N$ is the distribution of the noise $W_1, \ldots, W_N$. Then*

$$\inf_{\hat{\eta}} \sup_{(f,\eta,\mathbb{P}_N) \in \mathcal{F}} \mathbb{E}\left[\sup_{\mathcal{A}} (\hat{\eta}(\mathbf{X}) - \eta(\mathbf{X}))^2\right] \gtrsim \left(\frac{q}{N}\right)^{\frac{d+2}{d+1}} + N^{-\frac{2}{d+2}}, \tag{11}$$

*and*

$$\inf_{\hat{\eta}} \sup_{(f,\eta,\mathbb{P}_N) \in \mathcal{F}} \mathbb{E}\left[\sup_{\mathcal{A}}\sup_{\mathbf{x}} |\hat{\eta}(\mathbf{x}) - \eta(\mathbf{x})|\right] \gtrsim \left(\frac{q}{N}\right)^{\frac{1}{d+1}} + N^{-\frac{1}{d+2}}. \tag{12}$$

The proof is shown in section 4 in the supplementary material. In the right hand side of (11) and (12), $N^{-2/(d+2)}$ is the standard minimax lower bound for nonparametric estimation [29], which holds even if there are no adversarial samples. In the supplementary material, we only prove the lower bound with the first term in the right hand side of (11).

Compare Theorem 1, 2 and Theorem 3, we have the following findings. We claim that the upper and lower bound nearly match, if these two bounds match up to a polynomial of $\ln N$:

- From (10) and (12), with $h \sim \max\{(q/N)^{1/(d+1)}, N^{-1/(d+2)}\}$ and $T \sim \ln N$, the upper and minimax lower bound of $\ell_\infty$ error nearly match.

- If $q \lesssim \sqrt{N/\ln^2 N}$, from (9) and (11), let $h \sim N^{-\frac{2}{d+2}}$, the upper and minimax lower bound of $\ell_2$ match. In fact, in this case, the convergence rate of (5) is the same as ordinary kernel regression without adversarial samples, i.e. $h^2 + 1/(Nh^d)$. With optimal selection of $h$, the rate becomes $N^{-2/(d+2)}$, which is just the standard rate for nonparametric statistics [29, 38].

- The $\ell_2$ upper and lower bound no longer match if $q \gtrsim \sqrt{N/\ln^2 N}$. In this case, the optimal $h$ in (9) is $h \sim (q \ln N/N)^{2/(d+2)}$, and resulting $\ell_2$ error is $R_2 \lesssim (q \ln N/N)^{4/(d+2)}$, higher than the lower bound in (11).

This result indicates that the initial estimator (5) is optimal under $\ell_\infty$, or under $\ell_2$ with small $q$. However, under large number of adversarial samples, the $\ell_2$ error becomes suboptimal.

Now we provide an intuitive understanding of the suboptimality of $\ell_2$ risk with large $q$ using a simple one dimensional example shown in Figure 1, with $N = 10000$, $h = 0.05$, $M = 3$, $f(x) = 1$ for $x \in (0, 1)$, $\eta(x) = \sin(2\pi x)$, and the noise follows standard normal distribution $\mathcal{N}(0, 1)$. For each $x$, denote $q_h(x)$, $n_h(x)$ as the number of attacked samples and total samples within $(x - h, x + h)$, respectively. For robust mean estimation problems, the breakdown point is $1/2$ [39], which also holds locally for nonparametric regression problem. Hence, if $q_h(x)/n_h(x) > 1/2$, the estimator will collapse and return erroneous values even if we use Huber cost. In (a), $q = 500$, among which 250 attacked samples are around $x = 0.25$, while others are around $x = 0.75$. In this case, $q_h(x)/n_h(x) < 1/2$ over the whole support. The curve of estimated function is shown in Fig 1(b). The estimate with (5) is significantly better than kernel regression. Then we increase $q$ to 2000. In this case, $q_h(x)/n_h(x) > 1/2$ around 0.25 and 0.75 (Fig 1(c)), thus the estimate fails. The estimated function curve shows an undesirable spike (Fig 1(d)).

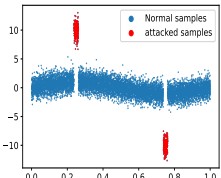 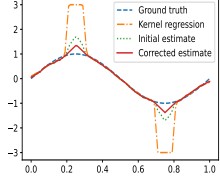 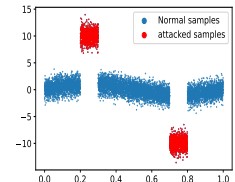 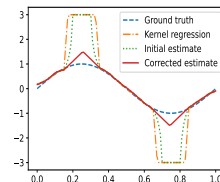

(a) Scatter plots with $q = 500$.

(b) Estimated results with $q = 500$.

(c) Scatter plots with $q = 2000$.

(d) Estimated results with $q = 2000$.

Figure 1: A simple example with $q = 500$ and $q = 2000$. In (a) and (c), red dots are attacked samples, while blue dots are normal samples. In (b) and (d), four curves correspond to ground truth $\eta$, the result of kernel regression, initial estimate and corrected estimate, respectively. With $q = 500$, the initial estimate (5) works well. However, with $q = 2000$, the initial estimate fails, while the corrected regression works well.

The above example shows that the best strategy for attacker is to focus on altering values at a small region. In this case, the local ratio of attacked samples surpasses the breakdown point, resulting in a wrong estimate. With such strategy and sufficient $q$, the initial estimator (5) fails to be optimal. Actually, (5) does not make full use of the continuity property of regression function $\eta$, and thus unable to detect and remove the spikes. A simple remedy is to increase $h$ so that $q_h(x)/n_h(x)$ becomes smaller. However, this solution will introduce additional bias. In the next section, we design a corrected estimator to improve (5), which will close the gap between upper and minimax lower bound with $q \gtrsim \sqrt{N/\ln^2 N}$.

## 4 Corrected Regression

In this section we propose and analyze a correction method to the initial estimator (5).

As has been discussed in section 3, the drawback of the initial estimator is that the continuity property of $\eta$ is not used. Consequently, an intuitive solution is to filter out the spike, and estimate $\eta$ here using

values in surrounding locations. Linear filter does not work here since the profile of the regression estimate will be blurred. Therefore, we propose a nonlinear filter as following. It conducts minimum correction (in $\ell_1$ sense) to the initial result $\hat{\eta}_0$, while ensuring that the corrected estimate is Lipschitz. Formally, given the initial estimate $\hat{\eta}_0(\mathbf{x})$, our method solves the following optimization problem

$$\hat{\eta}_c = \arg\min_{\|\nabla g\|_\infty \leq L} \|\hat{\eta}_0 - g\|_1, \tag{13}$$

in which

$$\|\nabla g\|_\infty = \max\left\{\left|\frac{\partial g}{\partial x_1}\right|, \ldots, \left|\frac{\partial g}{\partial x_d}\right|\right\}. \tag{14}$$

In section 5 in the supplementary material, we prove that the solution to the optimization problem (13) is unique.

(13) can be viewed as the projection of the output of initial estimator (5) into the space of Lipschitz function. Here we would like to explain intuitively why we use $\ell_1$ distance instead of other metrics in (13). Using the example in Fig.1(d) again, it can be observed that at the position of such spikes, $|\eta(\mathbf{x}) - g(\mathbf{x})|$ can be large. Other metrics such as $\ell_2$ distance impose large costs here, thus somewhat prevents the removal of spikes. Hence $\ell_1$ distance is preferred.

The estimation error of the corrected regression can be bounded by the following theorem.

**Theorem 4.** *(1) Under the same conditions as Theorem 1,*

$$\mathbb{E}\left[\sup_{\mathcal{A}} (\hat{\eta}_c(\mathbf{X}) - \eta(\mathbf{X}))^2\right] \lesssim \left(\frac{q \ln N}{N}\right)^{\frac{d+2}{d+1}} + h^2 + \frac{\ln N}{Nh^d}. \tag{15}$$

*(2) Under the same conditions as Theorem 2,*

$$\mathbb{E}\left[\sup_{\mathcal{A}}\sup_{\mathbf{x}} |\hat{\eta}_c(\mathbf{x}) - \eta(\mathbf{x})|\right] \lesssim \frac{Tq}{Nh^d} + h + \frac{\ln N}{\sqrt{Nh^d}}. \tag{16}$$

The proof is shown in section 6 in the supplementary material. Compared with Theorem 3, with $T \sim \ln N$ and a proper $h$, the upper and lower bound nearly match.

Now we discuss the practical implementation. (13) can not be calculated directly for a continuous function. Therefore, we find a approximate numerical solution instead. The detail of practical implementation is shown in section 1 in the supplementary material.

## 5   Numerical Examples

In this section we show some numerical experiments. In particular, we show the curve of the growth of mean square error over the attacked sample size $q$.

For each case, we generate $N = 10000$ training samples, with each sample follows uniform distribution in $[0, 1]^d$. The kernel function is

$$K(u) = 2 - |u|, \forall |u| \leq 1. \tag{17}$$

We compare the performance of kernel regression, the median-of-means method, initial estimate, and the corrected estimation under multiple attack strategies. For kernel regression, the output is $\max(\min(\hat{\eta}_{NW}, M), -M)$, in which $\hat{\eta}_{NW}$ is the simple kernel regression defined in (4). We truncate the result into $[-M, M]$ for a fair comparison with robust estimators. For the median-of-means method, we divide the training samples into 20 groups randomly, and then conduct kernel regression for each group and then find the median, i.e.

$$\hat{\eta}_{MoM} = \text{Clip}(\text{med}(\{\hat{\eta}_{NW}^{(1)}, \ldots, \hat{\eta}_{NW}^{(m)}\}), [-M, M]). \tag{18}$$

For the initial estimator (5), the parameters are $T = 1$ and $M = 3$. The corrected estimate uses (3) in the supplementary material. For $d = 1$, the grid count is $m = 50$. For $d = 2$, $m_1 = m_2 = 20$. Consider that the optimal bandwidth need to increase with the dimension, in (4), the bandwidths of all these four methods are set to be $h = 0.03$ for one dimensional distribution, and $h = 0.1$ for two dimensional case.

The attack strategies are designed as following. Let $q = 500k$ for $k = 0, 1, \ldots, 10$.

**Definition 1.** *There are three strategies, namely, random attack, one directional attack, and concentrated attack, which are defined as following:*

*(1) Random Attack. The attacker randomly select $q$ samples among the training data to attack. The value of each attacked sample is $-10$ or $10$ with equal probability;*

*(2) One directional Attack. The attacker randomly select $q$ samples among the training data to attack. The value of all attacked samples are $10$;*

*(3) Concentrated Attack. The attacker pick two random locations $\mathbf{c}_1$, $\mathbf{c}_2$ that are uniformly distributed in $[0,1]^d$. For $\lfloor q/2 \rfloor$ samples that are closest to $\mathbf{c}_1$, modify their values to $10$. For $\lfloor q/2 \rfloor$ samples that are closest to $\mathbf{c}_2$, modify their values to $-10$.*

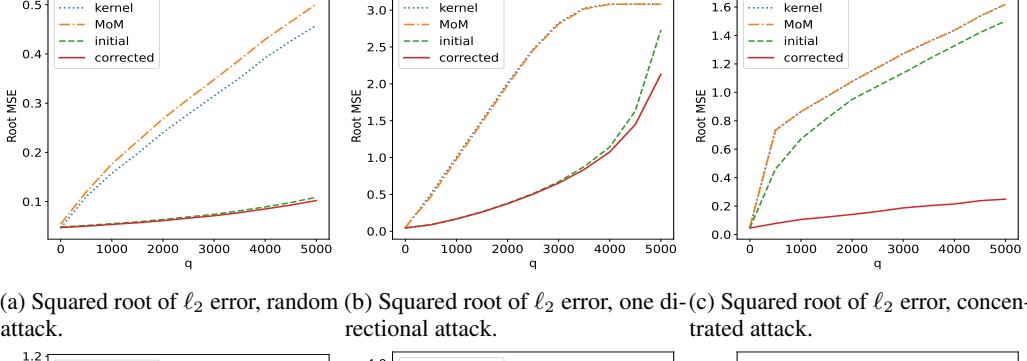

(a) Squared root of $\ell_2$ error, random attack.
(b) Squared root of $\ell_2$ error, one directional attack.
(c) Squared root of $\ell_2$ error, concentrated attack.

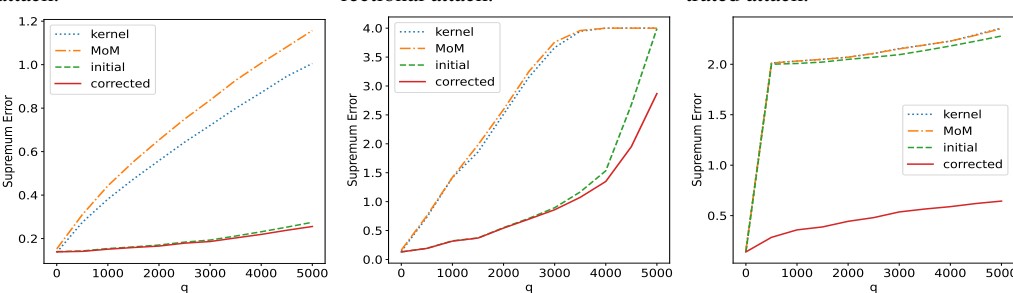

(d) $\ell_\infty$ error, random attack.
(e) $\ell_\infty$ error, one directional attack.
(f) $\ell_\infty$ error, concentrated attack.

Figure 2: Comparison of $\ell_2$ and $\ell_\infty$ error between various methods for one dimensional distribution.

For one dimensional distribution, let the ground truth be

$$\eta_1(x) = \sin(2\pi x). \tag{19}$$

For two dimensional distribution,

$$\eta(\mathbf{x}) = \sin(2\pi x_1) + \cos(2\pi x_2). \tag{20}$$

The noise follows standard Gaussian distribution $\mathcal{N}(0,1)$. The performances are evaluated using square root of $\ell_2$ error, as well as $\ell_\infty$ error. The results are shown in Figure 2 and 3 for one and two dimensional distributions, respectively. In these figures, each point is the average over 1000 independent trials.

Figure 2 and 3 show that the simple kernel regression (blue dotted line) fails under poisoning attack. The $\ell_2$ and $\ell_\infty$ error grows fast with the increase of $q$. Besides, traditional median-of-means does not improve over kernel regression. Moreover, the initial estimator (5) (orange dash-dot line) shows significantly better performance than kernel estimator under random and one directional attack, as are shown in Fig.2 and 3, (a), (b), (d), (e). However, if the attacked samples concentrate around some centers, then the initial estimator fails. Compared with kernel regression, there is some but limited improvement for (5). Finally, the corrected estimator (red solid line) performs well under all attack strategies. Under random attack, the corrected estimator performs nearly the same as initial one. For one directional attack, the corrected estimator performs better than the initial one with large $q$. Under concentrated attack, the correction shows significant improvement. These results are consistent with our theoretical analysis.

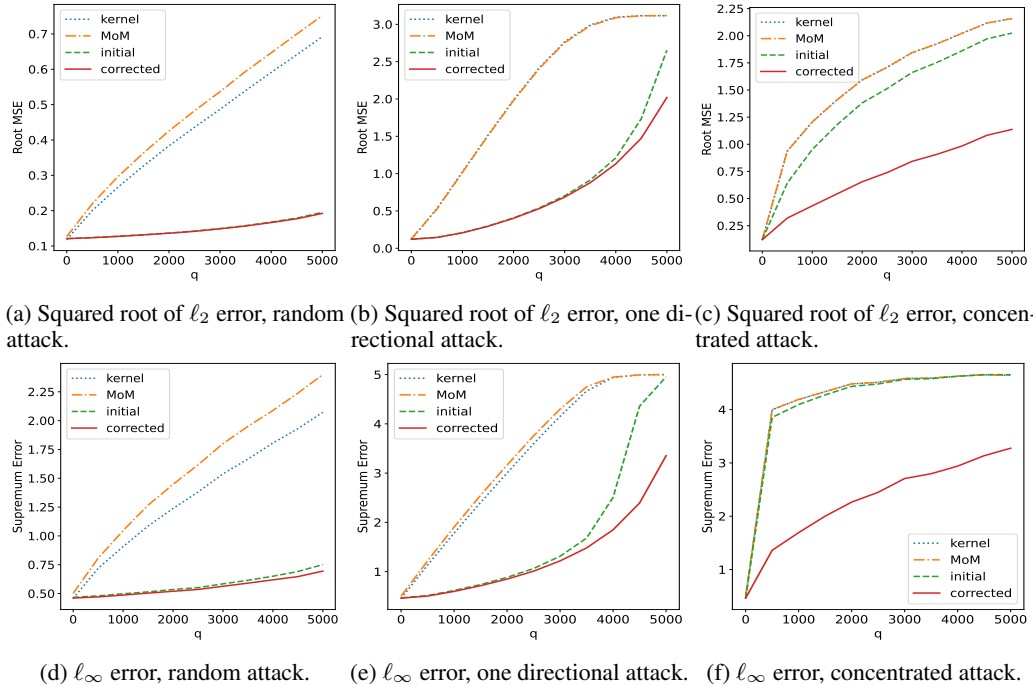

(a) Squared root of $\ell_2$ error, random attack.

(b) Squared root of $\ell_2$ error, one directional attack.

(c) Squared root of $\ell_2$ error, concentrated attack.

(d) $\ell_\infty$ error, random attack.

(e) $\ell_\infty$ error, one directional attack.

(f) $\ell_\infty$ error, concentrated attack.

Figure 3: Comparison of $\ell_2$ and $\ell_\infty$ error between various methods for one dimensional distribution.

## 6 Limitations

The major limitation is that for high dimensional feature distributions, the corrected estimator can be computationally expensive, since the number of grids grows exponentially with the dimensionality.

Moreover, our theoretical results rely on Assumption 1. Nevertheless, it is not hard to generalize these assumptions. For (a), we can use a local polynomial method to improve the convergence rate if $\eta$ satisfies higher order of smoothness. (b) limits the feature distribution. Actually, our analysis can be extended to heavy tail cases, in which the bandwidth can be made adaptive, such as [36, 37]. In order to achieve better tradeoff between bias and variance, in the regions with high pdf, bandwidth $h$ need to be smaller, and vice versa. Currently, we only focus on distributions without tails. (d) requires that the noise is sub-exponential. Such restriction can also be extended to the case in which the noise is only assumed to have bounded moments. In this case, we can let $T$ grow faster with $N$. Despite that we are convinced that all these assumptions can be extended with some modification, the current results focus on a simpler situation.

## 7 Conclusion

In this paper, we have provided a theoretical analysis of robust nonparametric regression problem under adversarial attack. In particular, we have derived the convergence rate of an M-estimator based on Huber loss minimization. We have also derived the information theoretic minimax lower bound, which is the underlying limit of robust nonparametric regression. The result shows that the initial estimator has minimax optimal $\ell_\infty$ risk. With $q \lesssim \sqrt{N/\ln^2 N}$, in which $q$ is the number of adversarial samples, $\ell_2$ risk is also optimal. However, for large $q$, the initial estimator becomes suboptimal. In particular, if the attacker focus their attack around some centers, then the resulting estimate shows some undesirable spikes at these centers. Actually, the drawback of initial estimator is that it does not make full use of the continuity of regression function, and hence unable to detect spikes and correct the estimate. Motivated by such discussion, we have proposed a correction technique, which is a nonlinear filter that projects the estimated function into the space of Lipschitz functions. Our theoretical analysis shows that the corrected estimator is minimax optimal even for large $q$. Numerical experiments validate our theoretical analysis.

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
