# OpenReview forum: "Robust Nonparametric Regression under Poisoning Attack"
_NeurIPS.cc/2023/Conference — Submitted to NeurIPS 2023_

### Official Review · Reviewer_QLoZ · 2023-07-05

**Soundness:** 2 fair
**Presentation:** 3 good
**Contribution:** 2 fair
**Rating:** 6
**Confidence:** 3

**Summary:**

The paper studies robust nonparametric regression under poisoning attacks. The input is samples from some fixed distribution and the goal is to approximate some unknown function on the input space. It is assumed that the values observed in q of the samples are adversarial. In this setting, classical approaches such as k-NN estimators can fail. This is because a single adversarial sample can affect the prediction on many points. In order to avoid this issue, the paper proposes robust variants of nonparametric regression.

The first result is a bound on the convergence rate of an M-estimator based on Huber loss minimization: the initial estimator has minimal optimal ell_infinity risk. When the number of adversarial samples is not too big, the ell_2 risk is optimal.

It is also shown that the estimator can suffer when many of the adversarial samples are concentrated within a small region. In order to resolve this issue, the paper proposes a correction step that projects the estimator into the space of Lipschitz functions. Upper bounds on the rate of convergence of the corrected estimator are established. Numerical results show that both estimators outperform standard methods in a simple low-dimensional synthetic scenario.



**Strengths:**

Nonparametric regression is a fundamental problem. The paper studies a natural adversarial setting where a small number of input samples are corrupted adversarially. This research direction can produce more robust useful data-analytic methods.

The upper and lower bounds match up to polylogarithmic factors when the number of adversarial samples is not too large.


**Weaknesses:**

The experimental evaluation is very limited. This is unclear whether the proposed methods can lead to significant robustness improvements in real-world scenarios.

No runtime analysis is given for the corrected estimator. This is a continuous optimization problem, so it is not clear how easy it is to solve in practice, especially in high or moderately-high dimensions.


**Questions:**

Assumption 1, c: The quantifier for x is missing.

In line 253, I don't quite understand the justification for using ell_1 norm in the optimization problem. Since ell_2 imposes a larger cost in the error region, it would seem that this makes it easier to remove spikes.

Definition 1 (3): One of the two floors should be a ceiling.



**Limitations:**

The authors have adequately addressed the limitations of their work.

---

> ### Author Rebuttal · Authors · 2023-08-02
>
> Thanks for this positive review.
>
>  **1.Nonparametric regression is a fundamental problem. The paper studies a natural adversarial setting where a small number of input samples are corrupted adversarially. This research direction can produce more robust useful data-analytic methods.**
>
> Reply: Thanks. The only thing is that "where a small number of input samples are corrupted adversarially" is not accurate. Actually, we allows a large number of adversarial samples, especially for the corrected estimator.
>
> This research can indeed be used as the background of future work, including (1) computational tractable nonparametric robust regression in high dimensions; (2) robust nonparametric classification and density estimation.
>
> **2.The experimental evaluation is very limited. This is unclear whether the proposed methods can lead to significant robustness improvements in real-world scenarios.**
>
> Reply: Thanks for this comment. We are now running more experiments in real data. There are actually no comparable baseline methods, since traditional methods are not designed for adversarial attack. Therefore, in our new experiments, we use other traditional robust nonparametric regression methods, including splines and additive models as baselines. The new method works much better compared with these traditional methods.
>
> It is straightforward for our method to outperform these baselines, since most of previous methods are designed for random noise instead of adversarial attack. Moreover, when the number of attacked samples is large, many previous methods fail.
>
> **3.No runtime analysis is given for the corrected estimator. This is a continuous optimization problem, so it is not clear how easy it is to solve in practice, especially in high or moderately-high dimensions.**
>
> Reply: In lines 141-143, we briefly analyzed the runtime of initial estimator, which is just $\ln(M/\epsilon)$ larger than kernel regression. For the corrected estimator, the time complexity is indeed larger, since the number of grids becomes large in high dimensions. This has been mentioned in lines 307-308.
>
> **4.Assumption 1, c: The quantifier for x is missing.**
>
> Reply: Thanks for this comment. We will add $\mathbf{x}\in \mathcal{X}$ after "For all $r<D$".
>
> **5.In line 253, I don't quite understand the justification for using $\ell_1$ norm in the optimization problem. Since $\ell_2$ imposes a larger cost in the error region, it would seem that this makes it easier to remove spikes.**
>
> Reply: $\ell_2$ imposes a larger cost in the error region, therefore, with $\ell_2$ error, the corrected estimate in the large error region tend to be closer to the initial estimate (which has spikes) to reduce $\ell_2$ cost, since the gradient here is large. What we hope is to make this region less affected by the spikes in the initial estimate. For this purpose, the gradient can not be too large at the large error region. As a result, $\ell_2$ error makes it harder to remove spikes. Therefore $\ell_1$ error is better.
>
> Put it in another way, the initial estimate has spikes, and we need to remove them. Therefore, the distance between the corrected estimate and the initial estimate is large. This large distance is reasonable, and our cost can not punish this distance too much. $\ell_2$ cost is proportional to the square of such distance, thus it prevents the correction. On the contrary, $\ell_1$ cost does not pose large cost for the distance between the initial estimate and the corrected one, hence the correction performance is much better.
>
> In the overall author rebuttal, we have included a figure to illustrate the effect of $\ell_1$ and $\ell_2$ loss with a simple example. Hope that this helps.
>
> **6.Definition 1 (3): One of the two floors should be a ceiling.**
>
> Reply: Setting one of two floors to be ceiling is OK, but not necessary. In line 94, we have mentioned that the attacker can manipulate up to q samples, i.e. $|\mathcal{B}|\leq q$. This means that $q$ is only an upper bound, and the attacker can decide to attack less than $q$ samples. Therefore, this is actually not a typo. We use floor in both two areas for symmetry. In our revision, we will add explanations to make it clearer.

---

### Official Review · Reviewer_i98Z · 2023-07-06

**Soundness:** 3 good
**Presentation:** 2 fair
**Contribution:** 2 fair
**Rating:** 5
**Confidence:** 3

**Summary:**

A robust version of the classic non-parametric problem is studied, when the training data is under an adversarial poisoning attack. The work is primarily theoretical, and makes assumptions on Lipschitzness and boundedness of the underlying function, boundedness of the data density, restrictions on sharpness of the data domain boundary, sub exponential noise. With some restrictions on the kernel, and appropriate parameters, upper bounds are shown on the $\ell_2$ and $\ell_\infty$ loss for the robust regression fit using Huber loss minimization. Lower bounds are obtained under the same assumptions, which match asymptotically with the upper bounds for $\ell_\infty$ and a correction is presented to match the upper bound for $\ell_2$ loss as well. Numerical simulations show usefulness of the algorithms on simulated one and two dimensional data.

**Strengths:**

1. The authors study robust nonparametric regression under poisoning attacks for smooth underlying functions under certain assumptions on the data and domain, under which asymptotically tight rates are obtained for $\ell_\infty$ loss by suitably parameterized Huber loss.
2. A novel correction method is proposed for which tight rates are obtained for $\ell_2$ loss as well. Together with above, the optimal algorithms under the given assumptions are obtained for both losses.
3. Numerical simulations are performed to verify the theoretical results.

**Weaknesses:**

1. The assumption needed for proving the theoretical result, specifically assumption 1(b) is too strong. Effectively, it reduces the possible test distributions to "near-uniform" distributions since there is both an upper and a lower bound on the probability density function. This assumption is too strong to make the results useful in most practical situations.
2. No insights is provided into the proofs of the theoretical results; in particular there are no proof sketches in the main body. This makes it challenging to understand the contributions, given the work is primarily theoretical.
3. Experiments only involve very specific numerical simulations. How does the approach work on real regression datasets? The effect of changing the hyperparameters is not studied and how they are selected is not described.
4. Huber loss is already known as a technique for robust nonparametric regression [1] and is not novel, the paper should mention that only the analysis under the present assumptions is new. The novel correction method is computationally intractable.

[1] Maronna, R. A., Martin, R. D., Yohai, V. J., & Salibián-Barrera, M. (2019). Robust statistics: theory and methods (with R). John Wiley & Sons.

**Questions:**

1. What is the intuitive behind needing assumption (c)? Do sharp corners make the model more susceptible to attack by a poisoning adversary?
2. Line 190-191 looks like notational overload for $h$?
3. How do T/M/h settings in the numerical simulations compare to theory?

**Limitations:**

The authors note several limitations of their work in a dedicated section.

---

> ### Author Rebuttal · Authors · 2023-08-02
>
> Thanks for the comments. We believe that for theoretical works, even if the method is proposed much earlier, new theoretical analysis is still a meaningful contribution, especially if the minimax lower bound is provided and the upper and lower bound matches. Examples include ref. [1-3] in this rebuttal, which are published in top journals.
>
> **1.The assumption needed for proving the theoretical result, specifically assumption 1(b) is too strong. Effectively, it reduces the possible test distributions to "near-uniform" distributions since there is both an upper and a lower bound on the probability density function. This assumption is too strong to make the results useful in most practical situations.**
>
> To the best of our knowledge, our work is the first attempt to solve robust nonparametric regression problem under adversarial attack. We think that it is reasonable to make a relatively simple assumption. Although it is somewhat not realistic, we need to avoid the readers to be distracted from complex assumptions.
>
> 'Near-uniform' assumption is commonly made in previous works even with clean data. In existing literatures. It is usually called "strong density assumption":
>
> [1]J.Audibert et al. "Fast learning rates for plug-in classifiers." Annals of Statistics 2007.
>
> [2]S.Gadat et al. "Classification in general finite dimensional spaces with the k-nearest neighbor rule." Annals of Statistics 2016.
>
> In these works, even though there is no attack, 'near-uniform' assumption is still needed. Therefore, the 'near-uniform' assumption is quite natural.
>
> Here we briefly discuss what will happen without the 'near-uniform' assumption. It is possible to make $\ell_2$ error converge, but the convergence rate is not optimal. To make it optimal, we need some adaptivity techniques, such as
>
> [3]P.Zhao, L.Lai. "Minimax rate optimal adaptive nearest neighbor classification and regression." IEEE Transactions on Information Theory 2021.
>
> Such adaptivity rules are far beyond the scope of this paper. The space is limited, and we would like to discuss further during the discussion period.
>
> **2.No insights is provided into the proofs of the theoretical results; in particular there are no proof sketches in the main body. This makes it challenging to understand the contributions, given the work is primarily theoretical.**
>
> Reply: We have provided insights in page 6, the end of section 3.
>
> **3.Experiments only involve very specific numerical simulations. How does the approach work on real regression datasets? The effect of changing the hyperparameters is not studied and how they are selected is not described.**
>
> We will add proof sketches and more experiments in future works. Selection of hyperparameters is discussed in lines 274-278.
>
> **4,Huber loss is already known as a technique for robust nonparametric regression [1] and is not novel, the paper should mention that only the analysis under the present assumptions is new. The novel correction method is computationally intractable.**
>
> (1) Novelty. "only the analysis under the present assumptions is new" is not accurate.
>
> Nonparametric regression has been studied much earlier. However, previous works on robust nonparametric regression focus on the case with (1) no corrupted data, but clean data has heavy tails; (2) corruption is random.
>
> With the development of machine learning, adversarial attack received extensive attention, such that the adversary can inspect samples and replace some of them with arbitrary values they like. This scenario is not analyzed in traditional robust statistics about nonparametric regression. This is the motivation of our work: what will be the effect of adversarial contamination on nonparametric regression, and how to defend?
>
> (2) Actually, we have mentioned the fact that the reviewer suggests.
>
> In line 54-55, we have mentioned that "It was originally proposed in [28], but to the best of our knowledge, it was not analyzed under adversarial setting."
>
> In line 109-110, we have repeated the statements.
>
> In line 307-308, we have mentioned that "The major limitation is that for high dimensional feature distributions, the corrected estimator can be computationally expensive."
>
> The main purpose of this corrected estimator is to show that the minimax lower bound is achievable. If q is not very large, correction is not necessary.
>
> Since the space is limited here, we would like to explain more during the discussion period later.
>
> **5.What is the intuitive behind needing assumption (c)? Do sharp corners make the model more susceptible to attack by a poisoning adversary?**
>
> In line 158-160, we have mentioned that "(c) prevents the shape of the corner of the support from being too sharp. Without assumption (c), the samples around the corner may not be enough, and the attacker can just attack the corner of the support."
>
> The attacker can just modify the values around the corner. In regions around the corner, clean samples are not enough to combat with poisoned samples.
>
> **6.Line 190-191 looks like notational overload for $h$?**
>
> We are not sure about the meaning of 'notational overload'. We guess that the reviewer is questioning about the notation $B(\mathbf{x}, h)$. We agree that it should be $B_h(\mathbf{x})$.
>
> **7.How do T/M/h settings in the numerical simulations compare to theory?**
>
> (1) The setting of T in theorems are aimed that rigorous analysis. Practically, it it not necessary to let T satisfy Assumption 3(b) (line 177-178) exactly. Therefore, we just set T=1.
>
> (2) In theory, M is required to be larger than the supermum of |eta(x)|. Our setting is M=3, satisfying this requirement.
>
> (3) For h, the theoretical analysis requires h>ln^2 N/N. This is only the mimimum h such that the theoretical analysis holds. Practically, in order to achieve a good bias-variance-robustness tradeoff, h need to be much larger than this lower limit. In our numerical experiments, h=0.03 for d=1, and h=0.1 for d=2.

---

> > ### Comment · Reviewer_i98Z · 2023-08-19
> > **Reply to rebuttal**
> >
> > Thanks for the detailed response. In particular, thank you for answering my questions and highlighting the difference relative to prior work.
> >
> > Regarding the "strong density assumption", as I suspected it is a relatively _strong_ assumption to study. For example, Audibert et al. 2007 also study the "mild density assumption" and also obtain optimal learning rates under it, which is more interesting. It would be good to include a discussion of the related work (thanks for providing it!) on the "strong density assumption" and mention that going beyond this assumption (potentially using the adaptivity techniques you mention) would be interesting.
> >
> > Despite the relatively strong assumption, the techniques could still be interesting to researchers. However, the lack of any proof sketches makes it harder to glean the technical novelties. It would be great to include proof sketches and more experiments (e.g. including real regression data) in your revision.
> >
> > `We are not sure about the meaning of 'notational overload'. `
> >
> > Regarding the notational overload in lines 190-191, note that $h$ is also the bandwidth parameter in equation (5).

---

> > > ### Author Response · Authors · 2023-08-19
> > > **Thank you very much for your response! We have further reply.**
> > >
> > > Thank you very much for replying to our response! In particular, we sincerely appreciate that you read the related paper of Audibert's. We would like to provide further response.
> > >
> > > # Strong Density Assumption
> > >
> > > Our work is the first attempt on adversarial attack on nonparametric regression. **From our experience in this area, compared with other initial works for some problems in nonparametric statistics, our assumption is not strong.** It is common to impose strong density assumption first. Follow-up works will relax these assumptions. In our revision, we will provide more discussions. Going beyond this assumption is indeed interesting, but this requires long and tedious analysis that distract the reader from the main focus.
> > >
> > > **Example 1.** In Audibert et al. [1], although the authors provided the matching upper bound under mild density assumption, **the purpose is to show that this bound is information-theoretic achievable, instead of providing a practical method to achieve it.** In particular, the matching upper bound is provided in Theorem 4.3 in [1], but this bound is achieved by eq (4.1). Note that (4.1) takes minimum over **a bunch of estimators $\eta\in N_{\epsilon_n}$ in which $N_{\epsilon_n}$ is very large.** Thus this method is not practical.
> > >
> > > There are several follow-ups including [2], which gives a practical matched upper bound, but **with a much larger set of unlabeled samples** (i.e. semi-supervised learning). Such unlabeled set may not always exist. Even if this unlabeled set exists, the assumption that unlabeled samples follow the same distribution as labeled samples (i.e. labeling is completely random) is not realistic.
> > >
> > > This work is then followed by [3], which does not require these unlabeled samples.
> > >
> > > **Example 2.** Another related problem is nonparametric entropy estimation. Assumption 1(c) in [4] is equivalent to the strong density assumption (the proof of equivalence is shown in Appendix G-B in [5]). [4] only analyzes the case with strong density estimation. In follow-up work [5], the assumption on the lower bound of density is removed.
> > >
> > > **Example 3.** Assumption 2 in [6].
> > >
> > > **In our revision, we will definitely include discussions of the related work about other problems in nonparametric statistics, and mention that going beyond this assumption would be interesting.**
> > >
> > > # Proof sketches and more experiments
> > >
> > > NeurIPS submission requires 9 pages only for initial submission, thus the space is not enough. When accepted, there will be an additional content page. We will definitely include proof sketches and experiments on real data if accepted.
> > >
> > > # Notational Overload
> > >
> > > Thanks for clarification. **Throughout this paper, $h$ always refer to bandwidth**. There is no notational overload here. The estimation error is large if in $B_h(x)$, most of samples are attacked. Lines 190-191 defines $B_h(x)$ instead of $h$. In our revision, we will emphasize it to avoid misleading.
> > >
> > > Thanks again for the further response!
> > >
> > >
> > > Reference:
> > >
> > > [1] J.Audibert et al. "Fast learning rates for plug-in classifiers." Annals of Statistics 2007.
> > >
> > > [2] T. I. Cannings et al. "Local nearest neighbour classification with applications to semi-supervised learning." Annals of Statistics 2020.
> > >
> > > [3] P.Zhao et al. "Minimax rate optimal adaptive nearest neighbor classification and regression." IEEE Transactions on Information Theory 2021.
> > >
> > > [4] W.Gao et al. "Demystifying fixed $ k $-nearest neighbor information estimators." IEEE Transactions on Information Theory 2018.
> > >
> > > [5] P.Zhao et al. "Analysis of KNN information estimators for smooth distributions". IEEE Transactions on Information Theory 2019.
> > >
> > > [6] H.Jiang "Non-asymptotic uniform rates of consistency for k-nn regression." AAAI 2019.

---

> > > > ### Comment · Reviewer_i98Z · 2023-08-21
> > > >
> > > > Thanks for the additional discussion on related work on nonparametric statistics. Also thanks for clarifying the notation in $B_h(\mathbf{x})$, and yes it would be good to use a consistent notation here (e.g. avoid $B(\mathbf{x}, h)$).
> > > >
> > > > I have increased my score based on the above discussion.

---

> > > > > ### Author Response · Authors · 2023-08-21
> > > > > **Thank you very much!**
> > > > >
> > > > > We appreciate much for your valuable comments. In the revision, we have included related discussions in the paper. Moreover, we have corrected the notations such as $B_h(\mathbf{x})$.
> > > > >
> > > > > Thanks for the improvement of rating!

---

### Official Review · Reviewer_pyjJ · 2023-07-06

**Soundness:** 3 good
**Presentation:** 2 fair
**Contribution:** 3 good
**Rating:** 5
**Confidence:** 3

**Summary:**

The authors propose two new nonparametric regression methods that and study their resilience under poisoning attacks.  The baseline for comparing the performance of the proposed methods is kernel regression (aka Nadaraya-Watson estimator).  The proposed methods are two, since the initial idea that the authors have, turns out to be vulnerable to poisoning attacks that are concentrated in small regions and a large budget is spent there.  Hence, the authors arrive at a `corrected' estimator which behaves better and is closer to optimal behavior.


After rebuttal:

For the largest part I believe that the authors have provided sufficient clarifications to various issues that were raised and moreover are willing to integrate comments and clarifications that came up during the discussion period in the final version of the manuscript. Therefore, I am increasing my score from reject to borderline accept; I am also increasing the soundness from fair to good as well as the presentation from poor to fair.

**Strengths:**

+ New methods for regression that are resilient to poisoning attacks.
+ Both theoretical and practical results.

**Weaknesses:**

+ Missing a section with preliminaries or background knowledge, where notions and notation that the authors use in the paper is well-defined. For example, in the section for preliminaries you can give information such as:
   - define loss functions,
   - discuss kernel regression and separate the presentation from your initial estimator,
   - discuss notions that you use when it is unclear what these things are; e.g., "bandwidth".
   - define functions that you use without explanations at the moment; e.g., in (18) we see Clip and med; what are the arguments and what do these functions do?

+ References not in alphabetical order.

+ In line 52-54, you indicate that your approach is similar to a combination of an $\ell_1$ and $\ell_2$ loss functions. At that point in the text, I was expecting a comparison with "elastic nets"; you may want to consider adding a small comment, or consider rephrasing and avoid such expectations from the readers.

+ The order by which some things are presented should probably change. For example, in lines 115-124, I think you need to give a slightly better explanation so that the reader can get better intuition, and moreover, this discussion should come up before you lay out the equations of the method that you propose.

+ You cannot start sentences with as in lines 167 or 168. You could add a word like "Part" or "Parts" in the beginning and make it read more naturally even if it takes a bit more space.

+ Larger font size in Figure 1 is expected.

+ It would be nice to see some commentary on the bounds and argue how they compare against each other and potentially compared to other methods that you cite in the literature.

+ In general, you have no comparison with parametric methods, either in theory, or in the experiments. Alternatively, provide an explanation as to why you don't have such comparisons.

+ It is unfortunate and normally it should not be an issue, but the authors need help on writing a paper that has fewer spelling or expression mistakes. The additional problem here is that the authors also have a big appendix, which I suspect is written similarly along the main text. So, fixing the main text is not enough. The paper needs to be proofread by someone in its entirety, including the appendix of the authors.

I think that you have a very interesting story to tell, but the paper needs restructuring and better presentation of what you have accomplished.

**Questions:**

Q1. In line 152, you define $D$ but this parameter does not show up anywhere.

Q2. In line 243, what is a "linear filter"?

Q3. In line 276, what do you mean by "optimal bandwidth"?

Q4. In line 308, what do you mean by "number of grids"?

Q5. What is $m$ in (18)?

Q6. Why don't you have some comparisons with parametric methods?

---

> ### Author Rebuttal · Authors · 2023-08-04
>
> Thanks the reviewer for valuable comments. The space is quite limited here. We reply to these comments briefly. We are willing to discuss more during the discussion period later.
>
> **1.Missing a section with preliminaries or background knowledge**
>
> Thanks for this suggestion. Loss function, kernel regression are defined in section 2. For "bandwidth", we have clarified it in line 104. It is parameter h. The term "bandwidth" is commonly used in many related articles, such as
>
> https://en.wikipedia.org/wiki/Kernel_density_estimation.
>
> We agree that it would be better to move these parts to preliminary section.
>
> med is median. Clip means to clip the value, i.e. $Clip(x, [-M, M]) =min(max(x, -M), M)$.
>
> **2.References not in alphabetical order.**
>
>  We have went through the requirement in NeurIPS website, which says that "Any choice of citation style is acceptable as long as you are consistent". It seems that alphabetical order is not required. We list references in the order they appear in the text.
>
> **3.In line 52-54, you indicate that your approach is similar to a combination of an $\ell_1$ and $\ell_2$ loss functions. At that point in the text, I was expecting a comparison with "elastic nets"; you may want to consider adding a small comment, or consider rephrasing and avoid such expectations from the readers.**
>
> Elastic net means regularization using $\ell_2$ plus $\ell_1$ loss. We are discussing nonparametric regression, thus regularization is not needed. More importantly, Huber loss is not $\ell_2$ plus $\ell_1$. Instead, Huber loss is $\ell_2$ for small values and $\ell_1$ for large values, thus combines the advantage of $\ell_2$ and $\ell_1$ and achieves good tradeoff between consistency and robustness.
>
> **4.The order by which some things are presented should probably change. For example, in lines 115-124, I think you need to give a slightly better explanation so that the reader can get better intuition.**
>
> Lines 115-124 are about parameters $M$ and $T$. These parameters appear in the equations above. Therefore, we have to put the discussion after the equations of the proposed methods.
>
> **5. You cannot start sentences with as in lines 167 or 168. You could add a word like "Part" or "Parts" in the beginning and make it read more naturally even if it takes a bit more space.**
>
> **6. Larger font size in Figure 1 is expected.**
>
> Thanks for these comments.
>
> **7.It would be nice to see some commentary on the bounds and argue how they compare against each other and potentially compared to other methods that you cite in the literature.**
>
> (1) Commentary on the bounds: Line 206-216 are the comments on the initial estimator with Huber cost minimization. Line 258-259 are comments about the corrected estimator. For the corrected estimator, the convergence rate already matches the minimax lower bound, thus we do not provide more comments.
>
> (2) For comparison with other methods. To the best of our knowledge, our work is the first attempt to solve robust nonparametric regression problem with adversarial attack. Therefore, there are no comparable baselines. Robust nonparametric regression has been studied much earlier, but they focus on clean data with heavy tails, or just random corruptions. Adversarial attack is a relatively new emerging field in recent years, and was not analyzed for nonparametric regression problems before.
>
> **8. In general, you have no comparison with parametric methods, either in theory, or in the experiments. Alternatively, provide an explanation as to why you don't have such comparisons.**
>
> Our consideration is that, if parametric assumptions are correct, i.e. the ground truth lies in the hypothesis space of parametric models, then parametric models perform definitely better than nonparametric models. If the parametric assumptions are not correct, then nonparametric methods should be better. Therefore, we do not expect to get interesting findings by comparison with parametric methods.
>
> **Q1. In line 152, you define $D$ but this parameter does not show up anywhere.**
>
> In eq.(9), (10), (15) and (16), we use $\lesssim$ notation. We have clarified in line 185-186 that $a\lesssim b$ if there exists a constant $C$ such that $a\leq Cb$. In theoretical results (9), (10), (15) and (16), the constants are omitted. These constants are affected by $D$. The details are shown in the proofs in the supplementary material.
>
> **Q2. In line 243, what is a "linear filter"?**
>
> linear filter means that output is linear in input, i.e. an operator $F$ is linear if for any function $f_1$, $f_2$ and any scalars $\lambda_1$ and $\lambda_2$, $F[\lambda_1 f_1+\lambda_2 f_2] = \lambda_1 F[f_1] + \lambda_2 F[f_2]$. Alternatively, $F[f]$ is a convolution of $f$ with another function $K_F$. Such convolution can blur the regression estimate.
>
> **Q3. In line 276, what do you mean by "optimal bandwidth"?**
>
> Optimal bandwidth means the best $h$. In line 104, we have clarified that "$h$ is the bandwidth that ...".
> In line 277, we have emphasized it again.
>
> **Q4. In line 308, what do you mean by "number of grids"?**
>
> In the numerical implementation of the corrected estimator (section 1 in the supplementary material), we divide the support into grids. Along $k$-th dimension, the grid count is $m_k$, thus the total number of grids is $\Pi_{k=1}^d m_k$. It has been mentioned in line 275. In revision, we will emphasize it more.
>
> **Q5. What is $m$ in (18)?**
>
> The number of batches for calculating median of means.
>
> **Q6. Why don't you have some comparisons with parametric methods?**
>
> This has been replied in previous comments. The effect of parametric methods depend on whether the parametric assumption is correct. Since the ground truth is designed by us, if we use parametric methods to fit the ground truth, it is definitely better than nonparametric methods. However, such comparison can not yield interesting findings, since we use nonparametric methods only when we have little knowledge about the ground truth.

---

> > ### Comment · Reviewer_pyjJ · 2023-08-15
> > **Thank you for the response**
> >
> > **References not in alphabetical order** I have always interpreted the quoted text as not mixing citation styles (e.g., in some parts one says "Hasselmo et al. (1995), while in other parts says [3]). In the .tex file that is provided as an example by NeurIPS, indeed, [3] is cited first and all the references are in alphabetical order. So, alphabetical order seems to be the right approach; not the order by which you cite the papers in the main text.
> >
> > **Elastic net** All I am saying is that some clarification should appear in the text, which is non-existent in the current form.
> >
> > **Comparison with parametric methods** One could identify the cases/domains where parametric/non-parametric methods are more suitable on real-world datasets and nevertheless compare the methods to see how far off their estimates are. I think it is still interesting to understand how much the results differ in various cases and the extent to which various assumptions hold on real datasets.
> >
> > I appreciate all your responses.  Indeed, my opinion for the paper has improved, but I still cannot recommend acceptance.

---

> > > ### Author Response · Authors · 2023-08-16
> > > **Thanks for your further response**
> > >
> > > Thanks for your response. We will definitely take your comment into our consideration and revise our paper accordingly.
> > >
> > > **References in alphebetical order**
> > >
> > > We have came across the NeurIPS 2022 accepted paper list:
> > >
> > > https://papers.nips.cc/paper_files/paper/2022
> > >
> > > From the paper list, among the first five NeurIPS accepted papers in the previous year, three of them list references in the order of appearance, instead of in the alphabetical order:
> > >
> > > https://papers.nips.cc/paper_files/paper/2022/hash/002262941c9edfd472a79298b2ac5e17-Abstract-Conference.html
> > >
> > > https://papers.nips.cc/paper_files/paper/2022/hash/00295cede6e1600d344b5cd6d9fd4640-Abstract-Conference.html
> > >
> > > https://papers.nips.cc/paper_files/paper/2022/hash/003a96110b7134d678cb675c6aea6c7d-Abstract-Conference.html
> > >
> > > Therefore, we believe that the alphabetical order is not required.
> > >
> > > **Elastic net**
> > >
> > > Thanks. We understand that it would be better to add some clarification, in order to help readers that are not in this area to understand. Maybe we can change the statement. Instead of claiming "Huber cost is a combination of $\ell_1$ and $\ell_2$ cost", maybe it is better to change the statement to "Huber cost is $\ell_2$ cost exactly with small input, and near to $\ell_1$ cost with large input". This will help to avoid readers from thinking about elastic net. We will make corresponding revision.
> > >
> > > **Comparison with parametric methods**
> > >
> > > In lines 45-48, we have mentioned that "in many real scenarios, due to problem complexity or lack of prior knowledge, parametric models are not always available. Therefore, we hope to explore effective schemes to overcome the robustness issue of nonparametric regression."
> > >
> > > This implies that if we can provide an appropriate parametric assumption, then it would be better to use parametric model. In the cases without such prior knowledge, then nonparametric methods are better.
> > >
> > > Actually, pros and cons of parametric vs nonparametric statistics are widely discussed in literatures. We agree with the reviewer that it is helpful to add several sentences for clarification, which will be included in our revision. However, the detailed comparison is beyond the scope of our paper.

---

> > > > ### Comment · Reviewer_pyjJ · 2023-08-20
> > > >
> > > > I must say that I am happy with the responses that the authors have provided to the various reviews, as well as the fact that the authors are willing to do some extra work as it is required by the reviewers for the final version of the paper.  For this reason I will increase my score from reject to borderline accept.

---

> > > > > ### Author Response · Authors · 2023-08-21
> > > > > **Thank you very much!**
> > > > >
> > > > > We are grateful for the improvement of rating, as well as your valuable comments. We are making corresponding revisions.
> > > > >
> > > > > Thanks!

---

### Official Review · Reviewer_yg2j · 2023-07-06

**Soundness:** 3 good
**Presentation:** 3 good
**Contribution:** 3 good
**Rating:** 6
**Confidence:** 2

**Summary:**

This paper studies nonparametric regression, where an adversary can corrupt $q$ samples from the training set. The paper proposes robust estimators for this problem.

**Strengths:**


The paper studied the Huber loss minimization approach under adversarial noise, giving theoretical upper bounds and lower bounds.



**Weaknesses:**

The organization could be improved, e.g. the last two paragraphs of section 3 could conceivably be placed elsewhere.

Some aspects of this paper are not really my area, so I cannot provide many helpful comments.

**Questions:**

Equation 2 and 3, what is expectation taken over?

**Limitations:**

The author addressed limitations.

---

> ### Author Rebuttal · Authors · 2023-08-04
>
> Thanks the reviewer for this positive review.
>
> 1.Comment: (from the strengths) The paper studied the Huber loss minimization approach under adversarial noise, giving theoretical upper bounds and lower bounds.
>
> Reply: Studying the Huber loss minimization under adversarial noise corresponds to the first step. We do not only study Huber loss minimization. Since Huber loss minimization is not perfect (not optimal for large q), we design a new correction method by nonlinear filtering.
>
> Robust parametric regression under adversarial attack has been analyzed in previous works. However, to the best of our knowledge, robustness of nonparametric methods against adversary is not discussed before.
>
> 2.Comment: The organization could be improved, e.g. the last two paragraphs of section 3 could conceivably be placed elsewhere.
>
> Reply: The reason we place it at the end of section 3 is that these two paragraphs are necessary to motivate the correction technique by nonlinear filtering.
>
> 3.Comment: Equation 2 and 3, what is expectation taken over?
>
> Reply: In equation 2, the expectation is taken over samples $(X_1,Y_1)$, ..., $(X_N, Y_N)$,  as well as the support of X (note that in (2), we use capital X)
>
> In equation 3, the expectation is taken only over samples $(X_1,Y_1)$, ..., $(X_N, Y_N)$. (note that in (3), we use the lowercase x).
>
> Our notations are common in nonparametric statistics. For example, eq.(9) in the following paper writes the expectation in similar way:
>
> Puning Zhao and Lifeng Lai. Minimax rate optimal adaptive nearest neighbor classification and regression. IEEE Transactions on Information Theory, 2021, 67(5): 3155-3182.

---

> > ### Comment · Reviewer_yg2j · 2023-08-19
> >
> > Thanks for the response. I will keep my score for now.

---

### Author Rebuttal · Authors · 2023-08-08

Thanks for the effort of all reviewers. We have received many valuable comments. We have the following general responses:

**1.Clarification of the novelty.**

Robust nonparametric regression has been studied much earlier, such as the book [1] mentioned by the reviewer i98Z. However, early works on robust statistics focuses either on (1) no corrupted samples, but clean samples have heavy tailed distributions; (2) samples are corrupted in random way.

With the development of machine learning in recent years, adversarial attack, especially poisoning attack on the training data, has received much attention, for example, see [2]. It means that, the adversary can pick some samples, and replace them with anything they like. As a result, adversarial attack is much stronger than random corruption, and is thus hard to defend. Note that this also explains why our second contribution, i.e. the corrected estimator, is necessary. For two cases (1) and (2) in traditional robust statistics mentioned above, correction technique may not be necessary.

**2.Related Work.**

In this paper, we focus on adversarial attack on training samples. To the best of our knowledge, our work is the first attempt on robust nonparametric regression, such that samples can be altered in an arbitrary way. Therefore, there is actually no baselines for comparison of theoretical bounds.

For empirical evaluations, we are now running experiments comparing our result numerically with other traditional methods (spline, additive models), which shows desirable results.

Moreover, there are some recent related works in other areas, such as density estimation [3].


**3. More explanations about the assumptions we made.**

Assumption (a)-(d) are common in previous literatures on nonparametric classification and regression. For example, Assumption 1(b) follows [4]. Moreover, Assumption 2 in a recent ICML paper [5] are just the same as our assumption (b) and (c).


**4. Future revisions.**

We would like to revise our paper in the following aspects:

(1) We will provide more explanations about the background of the research.

(2) We will provide more explanations about our assumptions. For example, we assume that the density has a lower bound in its support. This is common in literatures about nonparametric classification/regression, even without attack.

(3) Several terms need to be clarified. We will also revise other details, such as those mentioned by reviewer pyjJ.

(4) Other necessary revisions according to the reviews and later discussions.

**5. pdf file append here.**

The pdf file is to respond the question of reviewer QLoZ, in which the adversary concentrate all attacked samples in a small region, and thus initial estimate has spikes. The performance of minimizing $\ell_1$ norm (blue line) is better. If we use $\ell_2$ norm instead, then the $\ell_2$ loss at the region of spike will be large. To minimize $\ell_2$ loss, the corrected estimate has to move upward here.

**References**

[1] Maronna, R. A., Martin, R. D., Yohai, V. J., & Salibián-Barrera, M. (2019). Robust statistics: theory and methods (with R). John Wiley & Sons.

[2] Z. Tian et al. A comprehensive survey on poisoning attacks and countermeasures in machine learning[J]. ACM Computing Surveys, 2022, 55(8): 1-35.

[3] P. Humbert. Robust kernel density estimation with median-of-means principle. International Conference on Machine Learning, 2022

[4] Audibert, Jean-Yves, and Alexandre B. Tsybakov. "Fast learning rates for plug-in classifiers." Annals of Statistics, 2007.

[5] H Jiang, A Rostamizadeh. Active Covering. International Conference on Machine Learning, 2021.

---

### Decision · Program_Chairs · 2023-09-21

**Decision:**

Reject

**Comment:**

This work studies nonparametric regression in a robust setting where an adversary can corrupt a fraction of the dataset. The authors analyze an M-estimator based on Huber loss minimization and additionally develop a computationally inefficient estimator that is more sample efficient. The paper was discussed extensively by the reviewers. Overall, the results are non-trivial, but the exponential dependence on the dimension in the runtime of their proposed estimator limits the applicability of this work. Moreover, the comparison to prior art is lacking. For example, the authors do not cite or provide any comparison to the extensive recent line of works on algorithmic robust statistics. Overall, this work (in its current form) appears to be slightly below the threshold for publication at NeurIPS.